# Compressive Behaviors of Thin-Walled Steel Tube Stub Columns Filled with Self-Compacting Concrete Containing Recycled Aggregate

**DOI:** 10.3390/ma16186088

**Published:** 2023-09-06

**Authors:** Yunyang Wang, Shengwei Sun, Liqing Zhang, Yandong Jia

**Affiliations:** 1School of Civil and Architecture Engineering, Hunan University of Arts and Science, Changde 415000, China; 2China Construction Second Engineering Bureau Co., Ltd., Beijing 100054, China; ssw5256081@163.com; 3School of Civil Engineering, Tongji University, Shanghai 200092, China; 4School of Civil Engineering and Architecture, East China Jiaotong University, Nanchang 330013, China; 5State Key Laboratory of Performance Monitoring and Protecting of Rail Transit Infrastructure, East China Jiaotong University, Nanchang 330013, China; 6School of Civil and Architecture Engineering, Liaoning University of Technology, Jinzhou 121000, China

**Keywords:** recycled coarse aggregate, thin-walled steel tubes, self-compacting concrete, mechanical behaviors, failure modes

## Abstract

Natural resources have been excessively consumed, and large amounts of construction wastes have been generated, owing to the fast development of civil industry, causing crucial environmental issues. Therefore, reusable construction waste fabricated into recycled concrete offers a good strategy to solve this issue. Thus, this article first develops thin-walled steel tubes stub columns filled with self-compacting concrete containing recycled coarse aggregate. Afterwards, the compressive behaviors of the columns when undergoing axial compression loading to failure are explored. Subsequently, the effect of types of self-compacting concrete and wall thickness on failure modes and the relationships between load and displacement/strain is discussed comprehensively. Moreover, models of load–displacement/strain behaviors are proposed. The results show that columns with identical wall thicknesses containing both natural and recycled coarse aggregate display similar failure modes, mainly presenting as local buckling and rupture. The shape of the load–displacement/strain curves for identical wall thicknesses are almost the same. Nevertheless, the maximum load and stiffness of columns containing recycled coarse aggregate are lower than those of columns containing natural coarse aggregate. Additionally, the maximum loads corresponding to wall thickness of 1.2 mm and 3.0 mm are decreased by 18.4% and 5.8%, respectively. Moreover, the proposed models can reasonably evaluate the relationships between load and displacement/strain. This paper demonstrates that thin-walled steel tubular columns containing recycled coarse aggregate present positive compressive behaviors and thus exhibit great potential for developing environmentally friendly and sustainable civil infrastructures.

## 1. Introduction

Natural resources have been excessively consumed and large amounts of construction wastes have been produced owing to the fast development of the civil industry, resulting in crucial issues regarding the environment [1,2,3,4]. Therefore, releasing the heavy burden on the environment and establishing a sustainable society is an urgent challenge. Thus, reusing construction wastes to prepare new concrete offers a good strategy for solving this issue [5,6,7]. After construction and demolition wastes undergo crushing, grading, and cleaning processes, recycled aggregate can be obtained; it presents good and stable mechanical behaviors [8]. Utilizing the recycled aggregate can ease the release of pollution into the environment and diminish natural aggregate consumption, as well as decrease the carbon footprint of civil industries [9,10]. Natural coarse aggregate can be substituted with recycled coarse aggregate and can be applicable in non-structural positions [11]. Moreover, recycled coarse aggregate also presents potential applications in bridges and roadways [10,12,13]. Additionally, concrete members containing recycled coarse aggregate present good durability [14].

Self-compacting concrete possesses good fluidity and filling capacity, and is also non-segregating. It can spread into complicated positions without vibration. In addition, the hardened properties of self-compacting concrete are similar to traditional vibrated concrete [15,16]. Therefore, self-compacting concrete can be applied in complicated conditions [17,18]. This concrete was first developed in Japan in the 1980s. Subsequently, it has been developed quickly and spread all over the word [17,19,20,21].

In recent years, making use of recycled aggregate to prepare self-compacting concrete has been appreciated as a good strategy to minimize construction and demolition wastes and protect the environment [15,22,23,24,25,26,27]. The mechanical behavior of self-compacting concrete containing recycled aggregates is close to that of traditional concrete [23]. Majeed et al. pointed out that self-compacting concretes with natural and recycled aggregate present the same workability. However, compared with the compressive strength of concrete containing natural aggregate, the compressive strength of concrete containing recycled aggregate is reduced [28]. The hardened performances of self-compacting concrete are reduced when the replacement rate of recycled aggregate is within the scope of 10 wt.%~40 wt.%[29]. However, it has also been demonstrated that self-compacting concrete containing 40 wt.% recycled aggregate displays the same mechanical properties as normal self-compacting concrete [30]. In addition, the mechanical behaviors of self-compacting concrete containing 100 wt.% recycled aggregate is close to that of normal concrete; however, flexural toughness and stiffness is decreased [31]. Additionally, the density of self-compacting concrete containing recycled aggregate is lower than that of normal self-compacting concrete [29].

Concrete-filled steel tube columns present high strength, favorable ductility, and fire resistance as structural members. Moreover, steel tube columns can serve as formwork when concreting. This can save costs and construction time [32]. Thin-walled steel tube columns are attractive in civil engineering due to the fast development of steel with excellent performance. High-performance steel can maintain outstanding performance while reducing cost when applied in concrete-filled steel tube columns [33,34]. Regarding the advantages, many investigations on concrete-filled thin-walled steel tube columns have been conducted [33,35,36,37,38,39,40,41,42]. Because of their higher slenderness ratios, thin-walled steel tube columns filled with concrete are more prone to buckling, and ductility is poor [39,43,44,45,46]. Le et al. demonstrated that the collaboration effect between steel tube columns and infill concrete is intensive [47]. Wang et al. pointed out that the decrease rate of the descent phase between the relation of the load and displacement decreases upon increasing the wall thickness of the steel tube column [48]. Wang et al. indicated that the ductility coefficient of thin-walled tubes filled with concrete is higher than 4.0 [49]. Yu et al. researched eccentric compression behaviors on steel tube columns filled with self-compacting concrete containing recycled coarse aggregate columns. They demonstrated that the strain increased with the increase in recycled coarse aggregate [50].

Previous studies have concentrated on using recycled aggregate in producing self-compacting concrete [19,51,52,53] and thin-walled steel tube columns filled with self-compacting concrete [48]. These studies mainly concentrated on topics such as (1) steel tube columns filled with concrete [54,55,56,57]; (2) steel tube columns filled with self-compacting concrete [58]; (3) steel tube columns with thin wall thickness filled with concrete [34,47,49,59,60,61,62]; (4) steel tube columns with thin wall thickness filled with self-compacting concrete [48,63,64]; (5) steel tube columns filled with recycled concrete [65,66,67,68,69]; (6) steel tube columns filled with self-compacting concrete containing recycled aggregate [50]; (7) steel tube columns with thin wall thickness filled with recycled concrete [70,71]. However, little comprehensive investigation has been conducted concerning the two aspects on the behavior of recycled aggregate in applications in self-compacting concrete as well as steel tube columns with thin wall thickness filled with self-compacting concrete. Therefore, the aim of this research is to evaluate the feasibility and promote design and application of steel tube columns with thin wall thickness filled with concrete containing recycled coarse aggregate in stub columns. That is the primary novel contribution of this paper.

Firstly, six thin-walled steel tube stub columns filled with self-compacting concrete containing recycled and natural coarse aggregate are manufactured. In addition, four of the columns contain recycled coarse aggregate with a replacement rate of 100%. The other two columns are filled with self-compacting concrete containing natural coarse aggregate. Moreover, wall thicknesses of 1.2 mm and 3.0 mm are adopted. Subsequently, the compressive load behaviors of the columns are explored. The effect of the wall thickness on axial compressive behaviors and failure modes is analyzed in depth. This paper aims to create steel tubes filled with concrete as sustainable construction members and offer guidance for designing and facilitating the application of thin-walled steel tube stub columns containing recycled coarse aggregate in civil industry.

The differences between this work and the other existing studies are as follows. (1) the thin-walled steel tubes filled with self-compacting concrete containing recycled coarse aggregate. In this paper, we adopt a 100% replacement rate for the self-compacting concrete; (2) wall thicknesses of 1.2 mm and 3.0 mm are considered as the key parameters; (3) failure modes and axial compressive mechanical behaviors of the thin-walled steel tube columns filled with self-compacting concrete containing recycled aggregate are described; (4) theoretical models of the load–displacement/strain relationships of the columns containing recycled aggregate are proposed.

## 2. Experiments

### 2.1. Specimen Fabrication

In this paper, four thin-walled steel tube stub columns filled with self-compacting concrete containing recycled coarse aggregate with a circular cross section are fabricated. Another two thin-walled steel tube columns containing natural coarse aggregate are fabricated as control specimens. Two key parameters, wall thicknesses of 1.2 mm and 3.0 mm and types of coarse aggregates, are considered. Herein, the three specimens with a wall thickness of 1.2 mm are designed as one group. Two of them contain recycled coarse aggregate, and another contains natural coarse aggregate. Similarly, the wall thickness of the specimens of another group is 3.0 mm. Two of them contain recycled coarse aggregate, and another contains natural coarse aggregate. The diameter and height of the columns are set as 140 mm and 420 mm, respectively. The numbers, dimension, and properties of materials are shown in Table 1, wherein, D, t, and Lare on behalf of the outer diameter, wall thickness, and height of the steel tubes, respectively. f_ck_ and E_c_ stand for strength and modulus of elasticity of concrete, respectively. They are measured according to the specification [72]. F_y_, μ_s_, and E_s_ represent the yield strength, Poisson’s ratio, and modulus of elasticity of steel tubes, respectively. They are tested according to specification [73]. Self-compacting concrete containing natural and recycled coarse aggregate is described as N and R, respectively, and shown in Table 1. Wall thicknesses of 1.2 mm and 3.0 mm are labeled as T1 and T2, respectively. The numbers 1 and 2 at the end are the specimen number.

Some studies on steel tubes filled with concrete have been conducted, and the dimensions D and L and concrete strength have been referenced. A summary of the parameters of tubes filled with concrete in previous studies is shown in Table 2. It can be seen from Table 2 that the ranges of D and L are 100 mm~219.1 mm and 300 mm~1908 mm, respectively. In addition, the range of L/D is 2.19~12.0. Therefore, the dimensions of samples of D and L are designated as 140 mm and 420 mm, respectively, in this paper. The design of the dimensions of samples should take into account the ranges of parameter, strength of concrete and steel tubes, and feasibility of the experiments comprehensively. Table 2 shows that the range of the concrete strength in the references is from 27.8 MPa~193.3 MPa, and most of the concrete strength is within the scope of 36.3 MPa~72.51 MPa. Moreover, the most commonly used concrete strength in civil infrastructures is approximately 50 MPa. Therefore, concrete strength around 55 MPa is used.

Table 3 presents the mix proportion of the self-compacting concrete. The two aggregates are indicated with N and R, respectively. The mix proportion of concrete containing recycled and natural coarse aggregate is selected considering factors in combination with references in previous studies, experimental results, and the design codes [74]. The mix proportions of self-compacting concrete in previous studies are summarized in Table 4.

Cement with a grade of 42.5 is adopted and it was purchased from Henan Yongan Cement Co., Ltd. (Gongyi, China). Table 5 shows the chemical properties of the cement. Natural river sand is used for fine aggregate. The specific gravity of fly ash is 2.1. The polycarboxylate superplasticizer has a water reducing rate of more than 30% and was purchased from Muhu Co., Ltd. (Beijing, China). Tap water is used in the mixture. Figure 1 displays photos of the cement, fly ash, river sand, and coarse aggregates, respectively. The recycled coarse aggregate is obtained after the processes of crushing, cleaning, and grading of road concrete wastes. The strength of the road demolition concrete wastes is about 30 MPa, corresponding to middle-grade-strength concrete. The nominal maximum size of coarse aggregates is 20 mm. The sieved method is in accordance with the specification of ASTM [75]. Figure 2 shows the sieve size curves of coarse aggregates. Table 6 presents properties of the aggregates.

**Table 1 materials-16-06088-t001:** Numbers, dimensions, and properties of materials of the columns.

Number	D × t × L	L/D	D/t	Properties of Concrete		Properties of Steel Tubes		
	(mm^3^)			f_ck_ (MPa)	E_c_ (GPa)	F_y_ (MPa)	E_s_ (GPa)	μ_s_
N-T1-1	140 × 1.2 × 420	3.00	116.7	65.8	30.1	345.0	181.0	0.30
R-T1-1	140 × 1.2 × 420	3.00	116.7	54.4	25.7	345.0	181.0	0.30
R-T1-2	140 × 1.2 × 420	3.00	116.7	54.4	25.7	345.0	181.0	0.30
N-T2-1	140 × 3.0 × 420	3.00	46.7	65.8	30.1	358.3	202.0	0.28
R-T2-1	140 × 3.0 × 420	3.00	46.7	54.4	25.7	358.3	202.0	0.28
R-T2-2	140 × 3.0 × 420	3.00	46.7	54.4	25.7	358.3	202.0	0.28

**Table 2 materials-16-06088-t002:** Parameters on tubes filled with concrete.

References	D (mm)	t (mm)	L (mm)	L/D	Concrete Strength (MPa)
[58]	159.0	3.14~4.68	477.0~1908.0	3.0~12.0	52.88~72.51
[76]	114.3	2.74~5.90	300.0~900.0	2.62~7.87	56.0~107.0
[77]	114.3~219.1	3.6~10.0	250.0~600.0	2.19~2.74	148.8~193.3
[50]	140.0	3.63	500.0~1500.0	3.57~10.71	27.8~49.5
[64]	100~200	3.0	300.0~600.0	3.0	48.2
[78]	165.0	2.72	510.0	3.1	46.9~64.1
[69]	133.0~140.0	2.64~4.66	400.0~420.0	3.0	36.9~52.9
[65]	138~170.6	2.79~2.86	420.0~510.0	3.0	36.3~40.0
[66]	114.0~219.0	2.19~2.86	340.0~657.0	3.0	29.7~33.1
[79]	120.0	3.0	360.0	3.0	42.1~43.3

**Table 3 materials-16-06088-t003:** Mix proportions of self-compacting concrete (unit: kg/m^3^).

Numbers	Water	Cement	Sand	NCA *	RCA *	Fly Ash	Superplasticizer
							(wt.%)
N	198.96	401.47	836.46	768.5	-	122.1	0.4
R	192.8	389.0	845.9	-	770.0	118.3	0.4

* RCA: recycled coarse aggregate.* NCA: natural coarse aggregate.

**Table 4 materials-16-06088-t004:** Summary on mix proportions of self-compacting concrete (unit: kg/m^3^).

References	Water	Cement	NFA *	RFA *	NCA *	RCA *	Fly Ash
[80]	205	342	860.5	-	0.0	860.5	-
					259.9	606.3	
					435	435	
					611.7	262.1	
			914		914	0.0	
[81]	185	463	850	0	786	-	132
			765	74			
			680	159			
			595	238			
[51]	180	340	695	0	-	895	200
			521	153			
			348	305			
			174	458			
			0	610			
[52]	176	440	595.37	172.95	832.6	-	110
			396.91	345.9	832.6		
			198.46	518.85	832.6		
			-	691.79	832.6		
[19]	167	400	900	300	530	-	-
			600	600			
			300	900			
			-	1200			
[82]	277	430	846	0	301	278	185
			635	193	301	278	
			423	386	301	278	
			846	-	-	556	
[83]	190	475	900	-	600	120	75
[26]	194	315	738	-	596	149	135
					447	298	
					298	447	
[84]	277	430	820	-	326	326	185
				-	-	652	
[85]	171	427.5	765	-	749.2	-	-
			-	667	859.8	-	
			-	667	-	749.2	

* RFA: recycled fine aggregate. * NFA: natural fine aggregate. * RCA: recycled coarse aggregate. * NCA: natural coarse aggregate.

**Table 5 materials-16-06088-t005:** Chemical properties of the cement.

Properties	SiO_2_	Al_2_O_3_	Fe_2_O_3_	CaO	MgO	Na_2_O	SO_3_
Content (wt.%)	21.6	5.3	3.0	61.2	2.2	0.9	2.2

**Table 6 materials-16-06088-t006:** Physical properties of the aggregates.

Physical Properties	Specific Gravity	Porosity (%)	Crushing Index (%)	Fineness Modulus
River sand	2.7	41.8	-	2.46
Natural coarse aggregate	2.7	47.0	7.8	5.61
Recycled coarse aggregate	2.6	48.0	18.1	5.58

**Figure 1 materials-16-06088-f001:**
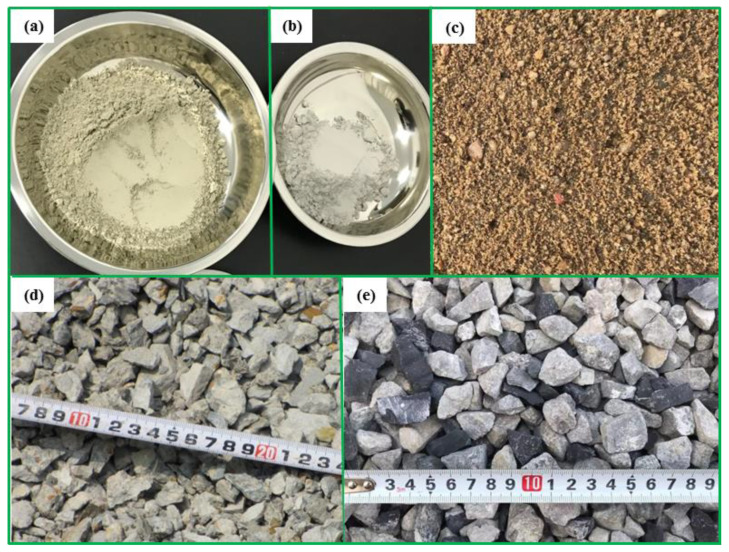
The photos of the materials. (**a**) Cement, (**b**) fly ash, (**c**) river sand, (**d**) recycled coarse aggregate, (**e**) natural coarse aggregate.

**Figure 2 materials-16-06088-f002:**
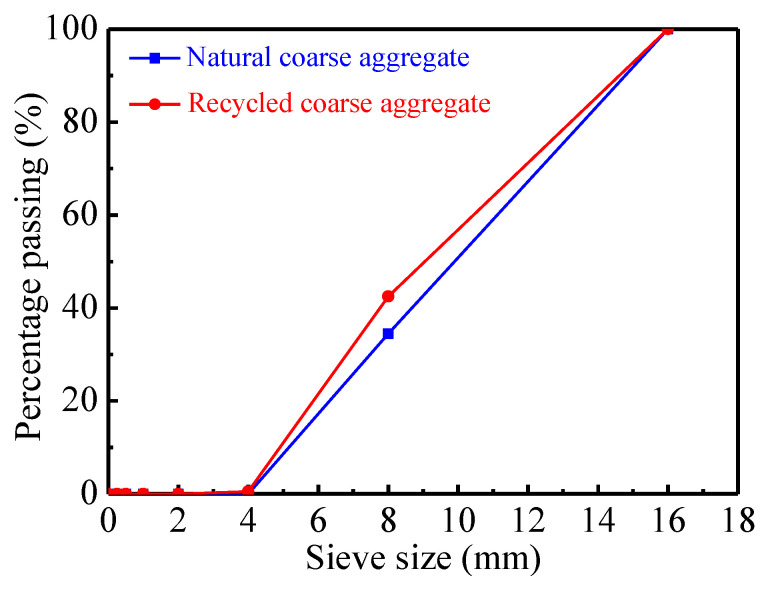
The sieve size curves of coarse aggregates.

Two square steel plates are welded on to the ends of the steel tubes. The side length and thickness of the square steel plates are 180 mm and 10 mm, respectively. The steel plate and steel tubes can work as a mold while casting concrete. The center of the steel plate and steel tubes are in a vertical line. The concrete is cast into the steel tubes slowly, and no external vibration is used. The columns are cured in an experiment room kept at 20 °C and 95% relative humidity for 28 d after concrete casting. However, about 2 mm of shrinkage of the infill concrete was generated after 28 d. For the sake of ensuring that the steel tubes and concrete are in the same plane, the uneven space between steel tubes and concrete is filled with cement paste. The top ends of the steel tubes are welded with another steel plate before testing. The typical manufacturing process of the columns can be seen in Figure 3.

### 2.2. Material Properties

The mechanical performance of the steel tubes, which are tested in accordance with the specification [86], are shown in Table 1. Tensile specimens are employed—three strips, which are cut from each type of the steel tubes—to test the mechanical performances.

The concrete used to fill the steel tubes and prepare the concrete specimen for measuring the mechanical properties is mixed together. Each type of concrete specimen is used to test the mechanical properties, including three cubes and six cylinders. After concrete casting, the concrete specimens and the columns are covered with plastic film and cured in the same conditions. Testing on the columns and concrete occurs at the same curing ages. Table 1 shows the mechanical properties of the concrete. Strain gauges are bonded on two opposite surfaces at the middle of the height of the concrete specimen during the testing process. The loading rate on both the cubes and cylinders of the concrete specimen is 0.20 mm/min.

### 2.3. Setup and Procedure

An axial compressive loading test of the columns is carried out after curing for 28 d using a universal testing machine. Figure 4 shows the photos of the loading process. As shown in Figure 4, two displacement meters are symmetrically arranged around the columns. Four strain gauges are bonded onto the steel tubes at the middle position along the axial direction to measure the axial strains, and they are numbered as 5 to 8. Similarly, another four strain gauges are bonded on the steel tubes in the middle position along the transverse direction to measure transverse strains, and they are numbered as 1 to 4. Figure 5 displays the schematic of the loading devices and locations.

Preliminary loading with a load amplitude of 20 kN and loading rate of 0.5 kN/min is carried out before formal testing. For the sake of ensuring that the columns are in the geometric center of the loading platform during the loading process, the location of the columns is adjusted carefully when the difference of the axial strains is more than 2% to guarantee that the column is in the center of the loading platform. The loading rate is applied with 0.5 mm/min in the formal loading stage. One-tenth of the calculated maximum load is set as the loading interval. In order to ensure the stability of the load and so that the collected data can more truthfully reflect the states, 3 min are maintained after each load interval is finished during the loading process. Loading is stopped after the load falls to 60 percent of the maximum load, and each column is tested for 2 h. Data of loads, displacements, and strains are collected automatically using a computer acquisition system during the loading process.

## 3. Results and Discussion

### 3.1. Failure Modes

Undergoing the axial compressive load, the columns present positive behaviors. Deformation along the axial direction changes approximately linearly with load. the ductility of the infill concrete has been enhanced owing to the confinement of the steel tubes. The columns present typical failure modes including local buckling and rupture. This characteristic is similar to high-strength thin-walled steel tubes filled with concrete columns. However, a failure slant angle close to 45° is generated on the infill concrete of the columns [60]. No obvious change has been generated on each of the columns until loading reaches 80% of the maximum load. However, a small amount of local buckling is generated on the surface near both ends of the columns, and this grows progressively worse when loading is close to the maximum load. The buckling and rupture are mainly distributed on the surface near the ends of the columns with a wall thickness of 1.2 mm, which is shown in Figure 6. Nevertheless, the buckling and rupture are distributed close to middle height of the columns in the columns with wall thickness of 3.0 mm. The infill concrete is crushed, and its volume is expanded at the failure stage. The thin-walled steel tubes’ inward buckling can be avoided with the expanded concrete. Therefore, the inner self-compacting concrete can work well with thin-walled steel tubes. Local buckling is also observed in high-strength steel tubes filled with self-compacting concrete columns. However, this phenomenon only occurred when withstanding peak load [58]. The reason that the generation of this local buckling of the high-strength steel tubes filled with self-compacting concrete columns in that work is later than in the present paper may due to the fact that the wall thickness and yield strength of their tubes are correspondingly thicker and higher than those in the present paper, respectively.

The buckling and rupture of the columns with identical wall thicknesses of steel tubes is similar to each other. Figure 6 shows the columns presenting similar failure modes after undergoing the maximum load. This phenomenon implies the columns with identical wall thickness containing natural and recycled coarse aggregate display similar failure modes. It also shows that recycled coarse aggregate presents great potential for application in steel tubes. However, the failure modes of the columns present small differences with different wall thickness. The D/t ratios of the steel tubes are 116.7 and 46.7 in this paper. Local buckling was generated earlier and developed faster and more extensively with a higher D/t ratio. This demonstrates that steel tubes with thicker wall thicknesses can exert a higher confinement on the concrete; thus, columns present less and more minor buckling. The steel tubes are ruptured when loading reaches the maximum load, and concrete is crushed, which is shown in Figure 6b. With continuous loading, the load decreases slowly with a wall thickness of 3.0 mm, while the load decreases rapidly with a wall thickness of 1.2 mm.

### 3.2. Relationship between Load and Displacement

The relationships between load and displacement of the columns subject to axial compression are displayed in Figure 7. The Ave-1 and Ave-2 in Figure 7 and Figure 8 represent the corresponding average values of specimens containing recycled coarse aggregate with wall thicknesses of 1.2 mm and 3.0 mm, respectively. Figure 7 demonstrates that the elastic stage of the columns is approximately 70% of the peak load. Both the strains in the axial and transverse directions are increased linearly upon increasing the load during this stage, which is shown in Figure 8. The ratio of the transverse strain to axial strain is close to 0.2, which indicates that the confinement by the steel tubes of the concrete is weaker. However, the elastic stage can increase to 90% of the peak load when using high-strength steel tubes [58]. As loading continues, the load is increased nonlinearly with the increasing of the displacement before the maximum load is reached. The relationship between load and displacement with a wall thickness of 3.0 mm presents a plastic plateau after the maximum load, while the load–displacement curves corresponding to the wall thickness of 1.2 mm descend suddenly. However, the load–displacement curves at the failure stage still present a horizontal tendency when the columns use high-strength steel tubes [60].

As shown in Figure 7a, the development shape of the load–displacement curves when the columns possess identical wall thicknesses are similar to each other. For example, the shape of the load–displacement curves of N-TI-1, R-T1-1, and R-T1-2 are almost identical to each other, except for the maximum load and the stiffness in the ascent phase of columns R-T1-1 and R-T1-2 being corresponding lower than that of N-TI-1. This is because of the infill self-compacting concrete of the column N-T1-1 using the natural coarse aggregate, while the columns R-T1-1 and R-T1-2 use the recycled coarse aggregate. The strength and modulus of elasticity of concrete containing natural coarse aggregate are 65.8 MPa and 30.1 GPa, respectively, which is, correspondingly, a little higher than that of the concrete containing recycled coarse aggregate: 54.4 MPa and 25.7 GPa, respectively. This phenomenon can also be observed between the columns N-T2-1, R-T2-1, and R-T1-2.

The theoretical model between load and displacement of the columns containing recycled coarse aggregate is established by fitting the load–displacement curve and shown in Figure 7a. Figure 7a shows that the R^2^ values of the two fitting curves are 0.9997 and 0.9993, respectively. The fitting curves coincide well with the experimental curves. This indicates that the suggested model offers a considerably accurate evaluation of the relationship between load and displacement. The theoretical model between load and displacement with different wall thicknesses of 1.2 mm and 3.0 mm is expressed as Equations (1) and (2), respectively.
(1)$$F=140.5\delta -108.2\delta^{2}+39.7\delta^{3}-3.9\delta^{4}$$
(2)$$F=-282.1\delta +251.2\delta^{2}-32.1\delta^{3}+0.9\delta^{4}$$
where F and δ are the axial load and displacement, respectively.

Figure 7b shows that the maximum loads of N-T1-1,R-T1-1, R-T1-2, N-T2-1, R-T2-1, and R-T2-2 are 1411.6 kN, 1105.4 kN, 1198.6 kN, 1637.6 kN, 1500.2 kN, and 1585.2 kN, respectively. The variation coefficients of the maximum load corresponding to different wall thicknesses of 1.2 mm and 3.0 mm are 0.057 and 0.039, respectively. Compared with the maximum load of column N-T1-1, the maximum loads of columns R-T1-1 and R-T1-2 are decreased by 21.7% and 15.1%, respectively. The average maximum load of R-T1-1 and R-T1-2 is decreased by 18.4%. It is implied that the maximum load of thin-walled steel tubes with wall thicknesses of 1.2 mm containing recycled coarse aggregate is lower than that of the same steel tubes containing natural coarse aggregate. This is mainly caused by the compressive strength of concrete containing natural coarse aggregate being higher than that of the concrete containing recycled coarse aggregate. Similarly, compared with the maximum load of column N-T2-1, the maximum loads of columns R-T2-1 and R-T2-2 are decreased by 8.4% and 3.2%, respectively. The average maximum load of R-T1-1 and R-T1-2 is decreased by 5.8%. It is implied that the maximum load of the thin-walled steel tubes with wall thicknesses of 3.0 mm containing recycled coarse aggregate is also lower than that of the same steel tubes containing natural coarse aggregate. The reasons for this are similar to those for the wall thickness of 1.2 mm.

Figure 7c shows displacement corresponding to the maximum load. Compared with the displacement of column N-T1-1, the displacements of columns R-T1-1 and R-T1-2 are increased by 24.9% and 15.1%, respectively. The average displacement of R-T1-1 and R-T1-2 is decreased by 19.8%. Similarly, compared with the displacement of column N-T2-1, the displacement of columns both R-T2-1 and R-T2-2 is decreased by 9.6%. This also demonstrates that the thicker wall thickness of the steel tubes contributes more significantly to improving the displacement compared to the improving effect of the concrete.

This demonstrates that the maximum load and ductility are improved with an increase in the wall thickness, while they are decreased with an increase in the D/t ratio. The maximum load and the displacement corresponding to the maximum load of the columns with wall thicknesses of 3.0 mm are 25.3% and 3.8% higher than those of the corresponding columns with wall thicknesses of 1.2 mm, respectively.

### 3.3. Relationship between Load and Strain

Figure 8 presents the relationship between the load and strain of columns undergoing compressive load. As shown in Figure 8, the load is approximately linearly varied with the axial and transverse strain until loading reaches 70% of the maximum load. Then, the load–strain curves become nonlinear, and strain is increased faster than load. The load–strain curves present a plastic plateau and then descend. Compared with the load–strain curves of N-T1-1, the load–strain curves of R-T1-1 and R-T1-2 display a similar development trend. However, the load–strain curves of N-T1-1 present better stiffness. Similarly, the curves of columns with different wall thicknesses present the same characteristics. This phenomenon also implies that the steel tubes with identical wall thicknesses containing recycled and natural coarse aggregate display similar relationships between load and strain. The reasons for this phenomenon are similar to the reasons for the recycled coarse aggregate’s effect on the maximum load of the columns.

Similar to the proposed model on load and displacement, a theoretical model of the relationship between the load and axial strain of the steel tube columns containing recycled coarse aggregate is also established by fitting the load–axial strain curve and is shown in Figure 8a. Figure 8a shows that the R^2^ values of the two fitting curves are 0.9997 and 0.9993, respectively. The fitting curves coincide well with test curves. This indicates that the suggested model offers a good evaluation of the relationship between load and axial strain. The theoretical model of the relationship between load and axial strain with wall thicknesses of 1.2 mm and 3.0 mm is expressed as Equations (3) and (4), respectively.
(3)$$F=-0.5\varepsilon -8.3\times 10^{-5}\varepsilon^{2}-3.9\times 10^{-9}\varepsilon^{3}+8.6\times 10^{-14}\varepsilon^{4}$$
(4)$$F=-0.5\varepsilon +2.9\times 10^{-5}\varepsilon^{2}+2.2\times 10^{-8}\varepsilon^{3}+1.8\times 10^{-12}\varepsilon^{4}$$
where F and ε represent the axial load and strain, respectively.

As shown in Figure 8b,c the axial/transverse strains corresponding to the maximum loads of N-T1-1,R-T1-1, R-T1-2, N-T2-1, R-T2-1, and R-T2-2 are −5700 με/5800 με, −7215.5 με/8035.6 με, −4526.2 με/4992 με, −6400 με/6200 με, −8410.3 με/5120 με, and −3709.9 με/8640 με, respectively. Compared with the axial/transverse strains of column N-T1-1, the axial/transverse strains of columns R-T1-1 and R-T1-2 are increased by 26.6%/38.5% and decreased by 29.3%/13.9%, respectively. The average axial/transverse strain of R-T1-1 and R-T1-2 is increased by 3.0%/12.3%. It is implied that the axial/transverse strain of steel tubes containing recycled coarse aggregate is higher than that of the average axial/transverse strain with identical steel tubes containing natural coarse aggregate. In addition, compared with the axial/transverse strains of column N-T2-1, the axial/transverse strains of columns R-T2-1 and R-T2-2 are increased by 31.4%/−17.4% and −42.0%/39.4%, respectively. The average axial/transverse strain of R-T2-1 and R-T2-2 is decreased by 5.3%/11.0%.

The average axial/transverse strains corresponding to the maximum load with steel tube wall thicknesses of 1.2 mm and 3.0 mm are 5870 με/6513 με and 6060 με/6880 με, respectively. Compared with the average axial/transverse strains of columns with a steel tube wall thickness of 1.2 mm, the axial and transverse strains corresponding to the maximum load with a wall thickness of 3.0 mm are increased by 3.1% and 5.3%, respectively.

Figure 8 shows that the strains in the axial direction increase faster than strains in the transverse direction before loading reaches the maximum load. The scope of the transverse strain to axial strain ratio is 0.23~0.36. However, the strains in the transverse direction are increased faster than the strains in axial direction after loading reaches the maximum load. The ratio of the strain in the transverse direction to the strain in the axial direction is more than 0.4. This indicates that the confinement by steel tubes of concrete is strong only after the maximum load is reached.

## 4. Conclusions

Thin-walled steel tube stub columns filled with self-compacting concrete containing recycled coarse aggregate are developed. The mechanical behaviors and failure modes of the columns undergoing axial compressive load to failure are investigated. The effects of wall thickness and types of self-compacting concrete on axial compressive behaviors are discussed. Theoretical models of the load–displacement/strain relationships are also proposed. Conclusions based on the results are as follows.

(1) The thin-walled steel tube columns with identical wall thickness filled with self-compacting concrete containing recycled and natural coarse aggregate display similar failure modes. In general, local buckling and rupture are the typical failure modes. However, the positions of the buckling and rupture are mainly distributed on the surface near the ends of the columns with a wall thickness of 1.2 mm, while they are distributed close to the middle height of the columns with a wall thickness of 3.0 mm. Moreover, the buckling and rupture are generated earlier and more extensively with the thinner wall thickness.

(2) The development trends of the relationships between load and displacement with identical wall thicknesses are similar to each other. Nevertheless, the maximum load and stiffness of the columns containing recycled coarse aggregate are lower than that of the maximum load and stiffness of the columns containing natural coarse aggregate. The load–displacement curves with thicker wall thickness present a longer plastic plateau after the maximum load, and better ductility.

(3) The maximum load of columns containing recycled coarse aggregate is correspondingly lower than the maximum load of the columns containing natural coarse aggregate. Compared with the maximum load of columns containing natural coarse aggregate, the maximum load of columns containing recycled coarse aggregate is decreased by 18.4% and 5.8%, corresponding to wall thicknesses of 1.2 mm and 3.0 mm, respectively. Similarly, the displacement corresponding to the maximum load of columns containing recycled coarse aggregate is increased by 19.8% and decreased by 9.6%, respectively.

(4) The models of the relationship between load and displacement/strain of the thin-walled steel tube columns containing recycled coarse aggregate subject to axial compression are established, and they present a good evaluation of load and displacement/strain.

(5) The columns with identical wall thickness filled with two different types of concrete display similar relationships between load and strain. However, the stiffness in the elastic stage of columns containing recycled coarse aggregate is lower than that of the stiffness in the elastic stage of columns containing natural coarse aggregate. Confinement by the steel tubes of the concrete is weaker before the maximum load, and it becomes more intensive after the maximum load.

This paper demonstrates that thin-walled steel tube columns containing recycled coarse aggregate present positive axial compressive behaviors and have great potential for applications in civil infrastructure, and promote the development of a renewable, sustainable, and carbon-neutral society. However, the results show that the maximum load, stiffness, and ductility of this column is a little lower than that of the corresponding values of the columns containing natural coarse aggregate. Moreover, the stability of the mechanical behaviors of the columns should also be explored comprehensively.

## Figures and Tables

**Figure 3 materials-16-06088-f003:**
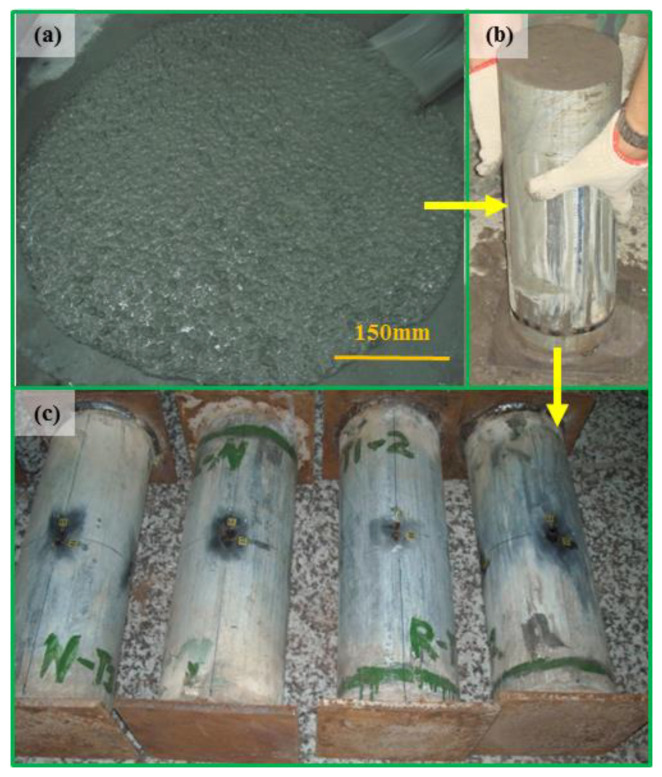
Manufacturing process of the columns. (**a**) The self-compacting concrete, (**b**) filling concrete into steel tubes, (**c**) the prepared columns.

**Figure 4 materials-16-06088-f004:**
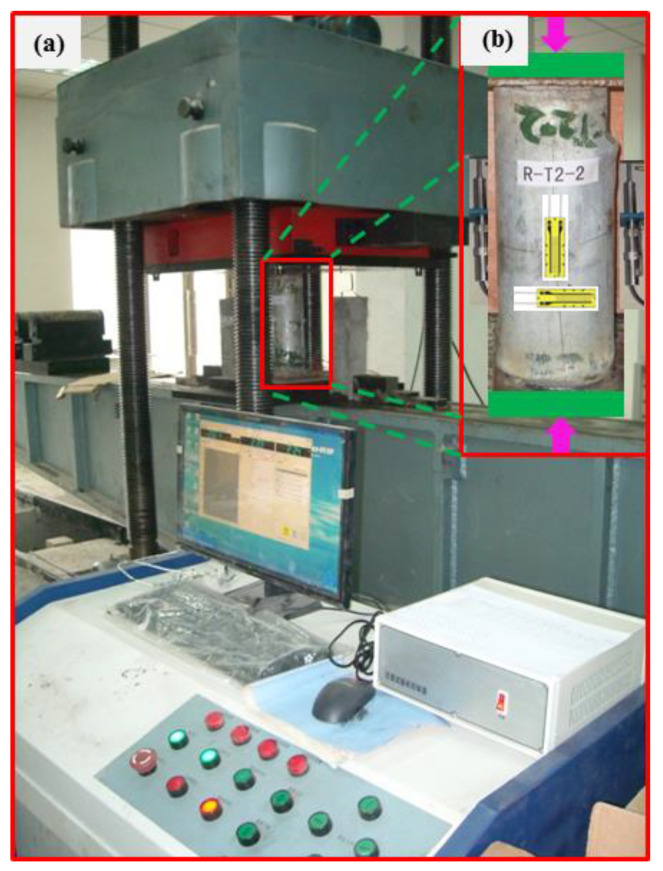
Photos of loading process of the columns. (**a**) The testing machine, (**b**) the columns subjected to compression.

**Figure 5 materials-16-06088-f005:**
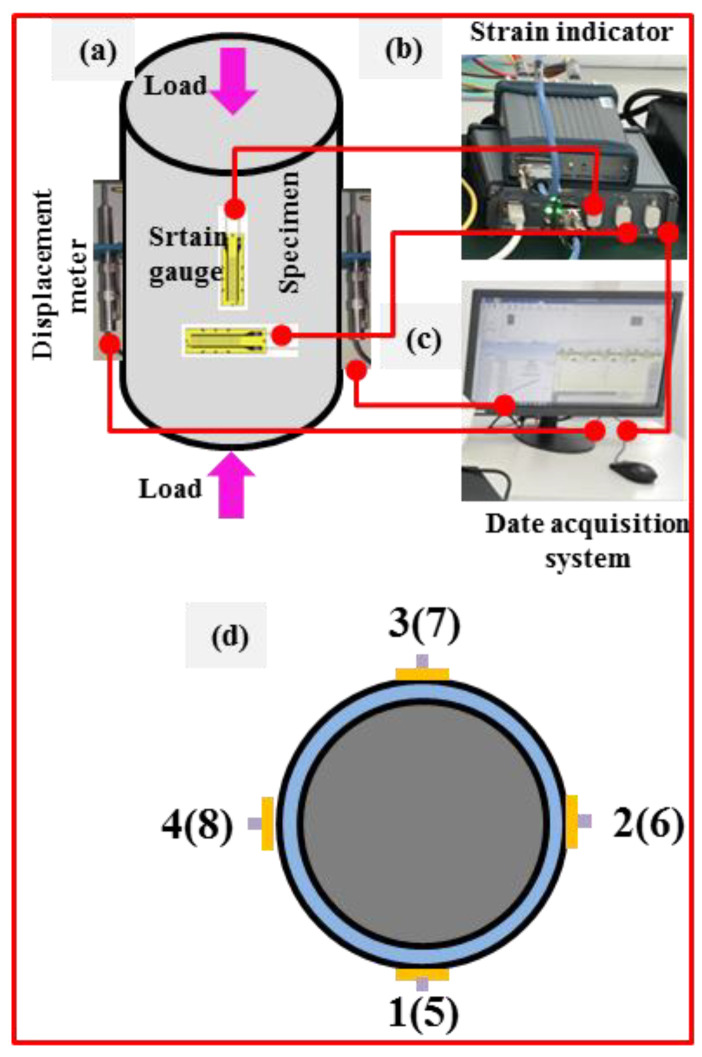
Schematic of loading devices and locations. (**a**) Schematic of the loading specimen, (**b**) strain indicator, (**c**) data acquisition system, (**d**) arrangement of the strain gauges on specimen section.

**Figure 6 materials-16-06088-f006:**
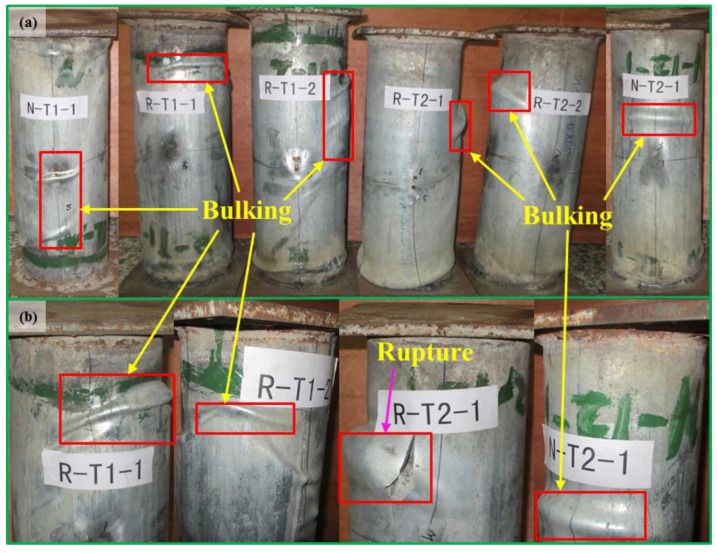
Failure modes. (**a**) The overall specimens, (**b**) the local positions.

**Figure 7 materials-16-06088-f007:**
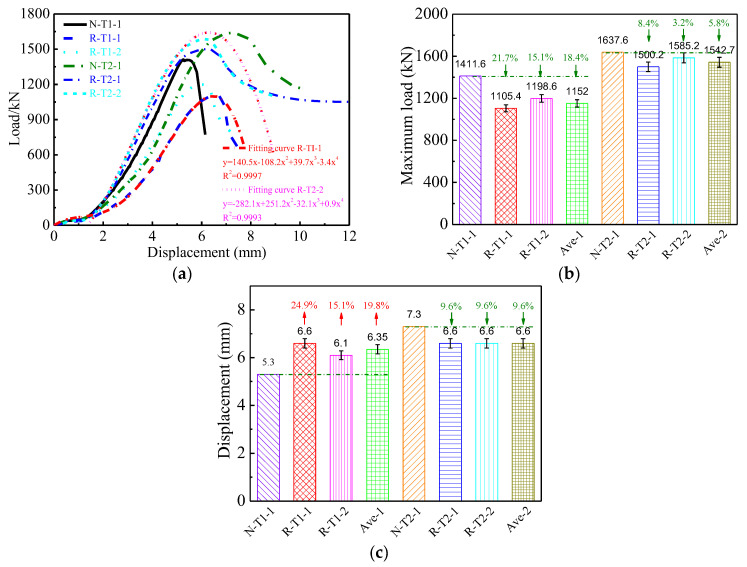
Relationship between loads and displacements. (**a**) Load–displacement curves, (**b**) the maximum loads, (**c**) displacements corresponding to the maximum loads.

**Figure 8 materials-16-06088-f008:**
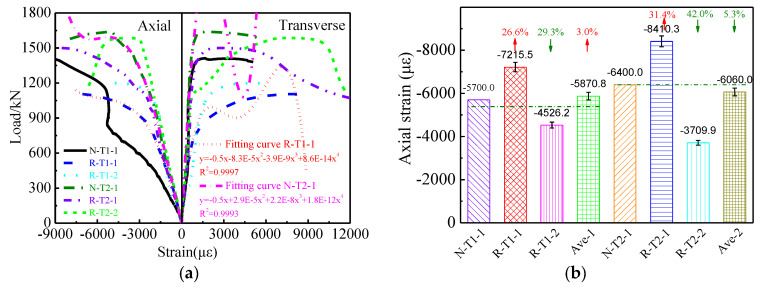
Relationship between load and strain. (**a**) Load–strain curves, (**b**) axial strain corresponding to the maximum load, (**c**) strain in transverse direction corresponding to the maximum load.

## Data Availability

Not applicable.

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
