# Peer review of "Compressive Behaviors of Thin-Walled Steel Tube Stub Columns Filled with Self-Compacting Concrete Containing Recycled Aggregate"

_materials, 2023, doi:10.3390/ma16186088_

Round 1

Reviewer 1 Report

The authors used recycled aggrate in steel tubes. Their performance under axail forces were investigated. The number of samples are very low. The conclusion is low and results are need to be imprvoment.

The abstract should contain important results of the study.

Novelty is not clear. Very same studies are already exists. What is the difference?

There are many types of recycled aggrate in concrete such as glass, marble, coal bottom ash, ceramic and etc. This can be indicated using following studies: use of recycled coal bottom ash in reinforced concrete beams as replacement for aggregate; mechanical behavior of crushed waste glass as replacement of aggregates;flexural behavior of reinforced concrete beams using waste marble powder towards application of sustainable concrete; improving bond performance of ribbed steel bars embedded in recycled aggregate concrete using steel mesh fabric confinement; effects of waste powder, fine and coarse marble agregates on concrete compressive strength;

How did you decide the dimensions of samples? 140mm and 420mm?

Why did you use high strength concrete around 55 MPa?

How this recycled materials for this study is obtained?

Add sieve analysis results in Figure.

What is chemical properties of cement

Give more detail on failure modes.

Result sections should be significanlty improved.

The reason for selecting design mixture should be added.

Compare your results with existing studies

Add photos for utilized materials. There is no photo related to which materials are utilized.

Add recent studies on this subject to introduction. There are many studies on the introduction for this topic.

Conclusion should be improved. The recommendation consdiering all test should be given for engineers.

Reviewer 2 Report

In the section of failure mode to add some analytical expressions for . Relationship between load and strain, so as to compare analytical expressions with these curves of figures 5 and 6.

no comments

Reviewer 3 Report

Overall, the manuscript presents an interesting study on the development of thin-walled steel tubular stub columns filled with self-compacting concrete containing recycled aggregate (TSTCFSCCR). However, there are several areas that require improvement to enhance the quality and clarity of the paper.

Paper Type:

The author(s) should explicitly state the type of paper this manuscript represents, whether it is an Article, Review, Communication, or any other appropriate category.

Abstract:

The abstract needs improvement as it currently duplicates information found in the introduction. To enhance its effectiveness, the abstract should provide a concise summary of the research's objective, methodology, key findings, and conclusions without repeating content from other sections.

Font Size:

Please adjust the font size in lines 36 and 104 to comply with the journal's formatting guidelines.

Citations:

In line 37, you mentioned previous studies on SCC containing recycled aggregate and concrete-filled thin-walled steel tubular columns. It is essential to provide proper citations for these previous works to strengthen the literature review and acknowledge prior research.

Originality:

The manuscript lacks clarity regarding the original contributions of this work. In the introduction, it is stated that this research explores the performance of thin-walled steel tubular stub columns filled with self-compacting concrete containing recycled aggregate. However, the specific aspects that make this work original and different from previous studies should be clearly highlighted.

Comparison with Non-Recycled Aggregate:

As the study involves testing recycled aggregate, it is crucial to include a comparison with the use of non-recycled aggregate. This comparison can help readers understand the potential benefits or drawbacks of incorporating recycled materials in this context.

Theoretical and Design Models:

To support the findings, theoretical and design models based on the samples should be provided. Including such models can enhance the understanding of the experimental results and their practical implications.

Figure Labels:

Ensure that all figures are appropriately labeled as per the magazine's guidelines. Clear and descriptive figure labels are essential for readers to comprehend the visuals easily.

In conclusion, the manuscript addresses an interesting topic, but it requires significant improvements in various aspects such as abstract clarity, proper citation of previous works, highlighting originality, incorporating comparisons, theoretical models, and adhering to figure labeling guidelines.

Font Size:

Please adjust the font size in lines 36 and 104 to comply with the journal's formatting guidelines.

Reviewer 4 Report

Article "Performances of thin-walled steel tubular stub column filled with self-compacting concrete contains recycled aggregate under axial compression" is devoted to the thin-walled steel tubular stub column filled with concrete wastes. The article is suitable for this journal in terms of subject matter. However, the following questions and comments were found during the study of this article:

- The paper has a very small introduction, it would be useful to expand and complete it.

- Line 36 - "Thin-walled;" bolded;

- line - it is necessary to make the font the same;

- there are no dots after figure numbers;

- The font of "Acknowledgements" is also different from that of the rest of the text.

- It would be helpful for the authors to point out the weaknesses of this paper as well as possible ways to continue it in the future.

Round 2

Reviewer 1 Report

The paper has been significantly imrpoved. There are still unsolved issues. After this, the paper can be accepted:

Please give more details for comment 6. What was the strength of recycled conrete. Is it dirved from low strength or high strength concrete.

Please readdress comment 4 for inlcuding the suggestions

Comment 8: Failure modes can be shown in Figures with highlighting

Reviewer 3 Report

The authors have made enhancements to the article's overall presentation, significantly improving its quality. The revisions undertaken are satisfactory, rendering the article suitable for publication. Just a few minor adjustments remain:

Please ensure the correct font size is applied throughout the article. On line 159, it is stated that "Table 3 presents the mix proportion of aggregate."

In addition, I recommend adding a personalized perspective to the conclusion section. Express your own viewpoint on the direction in which research in this field is likely to progress. Tailor your insights to the theme of the "Materials for Carbon-Neutral Infrastructures" issue, which serves as the context for publication. Your perspective will contribute to the depth and relevance of the article within the specific framework of this magazine.

Please ensure the correct font size is applied throughout the article. On line 159, it is stated that "Table 3 presents the mix proportion of aggregate."

Reviewer 4 Report

The authors have responsibly approached to the correction of the article, so the article can be accepted for publication in the current form
